# Identification of Sesamin from *Sesamum indicum* as a Potent Antifungal Agent Using an Integrated in Silico and Biological Screening Platform

**DOI:** 10.3390/molecules28124658

**Published:** 2023-06-09

**Authors:** Khushbu Wadhwa, Hardeep Kaur, Neha Kapoor, Simone Brogi

**Affiliations:** 1Fungal Biology Laboratory, Ramjas College, University of Delhi, Delhi 110007, India; 2Department of Chemistry, Hindu College, University of Delhi, Delhi 110007, India; 3Department of Pharmacy, University of Pisa, Via Bonanno 6, 56126 Pisa, Italy

**Keywords:** virtual screening, computer-aided drug discovery, antifungal agents, *Candida* species, *exo*-1,3-β-glucanase, synergism

## Abstract

Due to the limited availability of antifungal drugs, their relevant side effects and considering the insurgence of drug-resistant strains, novel antifungal agents are urgently needed. To identify such agents, we have developed an integrated computational and biological screening platform. We have considered a promising drug target in antifungal drug discovery (*exo*-1,3-β-glucanase) and a phytochemical library composed of bioactive natural products was used. These products were computationally screened against the selected target using molecular docking and molecular dynamics techniques along with the evaluation of drug-like profile. We selected sesamin as the most promising phytochemical endowed with a potential antifungal profile and satisfactory drug-like properties. Sesamin was submitted to a preliminary biological evaluation to test its capability to inhibit the growth of several *Candida* species by calculating the MIC/MFC and conducting synergistic experiments with the marketed drug fluconazole. Following the screening protocol, we identified sesamin as a potential *exo*-1,3-β-glucanase inhibitor, with relevant potency in inhibiting the growth of *Candida* species in a dose-dependent manner (MIC and MFC of 16 and 32 µg/mL, respectively). Furthermore, the combination of sesamin with fluconazole highlighted relevant synergistic effects. The described screening protocol revealed the natural product sesamin as a potential novel antifungal agent, showing an interesting predicted pharmacological profile, paving the way to the development of innovative therapeutics against fungal infections. Notably, our screening protocol can be helpful in antifungal drug discovery.

## 1. Introduction

*Candida* is an opportunistic fungal pathogen that causes a substantial amount of mortality worldwide in immunocompromised patients suffering from diabetes, cancer, AIDS, organ transplantation, and other immunosuppressive conditions. *Candida* is the most predominant fungal pathogen involved in both superficial and systemic infections and is the third most commonly isolated bloodstream pathogen in hospitalized patients [1]. *Candida albicans* and emerging non-*albicans Candida* (NAC) species such as *C. glabrata*, *C. parapsilosis*, *C. tropicalis*, and *C. krusei* can cause superficial infections of the oral and vaginal mucosa, as well as disseminated bloodstream and deep-tissue infections [2]. Antifungal agents such as azoles and polyenes exhibit serious side effects and toxicity in the host; thus, there is an urgent need to explore novel antifungal agents. Currently, limited antifungal drugs are available for the treatment of fungal infections, while the emergence of drug-resistant strains is increasing [3]. A promising drug target in antifungal drug discovery is represented by *exo*-1,3-β-glucanase [4]. This enzyme is involved in cell wall remodeling of *Candida* spp. [5]. The cell wall of *Candida* spp. is highly dynamic in nature and forms the outermost layer of the cell. It helps to maintain the shape and integrity of the cell and forms various interactions with host cells [6]. The cell wall of *Candida* spp. is mainly composed of polysaccharides, glucan, chitin, and mannan, which form the outer fibrillar layer. The central core of the cell wall is made up of a polymer of β-1,3-glucan, which is a branched glucan with β-1,6 branch points to which chitin and galactomannan are covalently attached. Mannans are often associated with proteins and lipids and represent 30–35% of total cell wall polysaccharides [7,8]. Cell wall protein function during cell wall assembly is crucial for remodeling and adhesion to the host cell, abiotic surface, biofilm formation, invasion of epithelial tissues, and is also pivotal in escaping the mechanism of host immune systems. In addition to this, the cell wall contains various enzymes, including chitinases, glucanases, peptidases, and glycosyltransferases that are generally involved in the remodeling of the cell wall, thus providing flexibility and mechanical strength during cell growth or lysis in response to oxidative stress [8]. β-1,3-glucan and β-1,6-glucan are the major components of the cell wall and account for 40% and 20% of total component, respectively. β-1,3-glucan hydrolyzing enzymes are classified into two types, such as *exo*-1,3-β-glucanases and *endo*-1,3-β-glucanases. *endo*-1,3-β-Glucanase cleaves inside a glucan chain to release small oligosaccharides, while *exo*-1,3-β-glucanase releases glucose monomers from the non-reducing end as a sole product [5]. In *C. albicans*, three related *exo*-1,3-β-glucanases, namely, Xog1p, Exg2p, and Spr1p, hydrolytically remove glucose from the ends of cell wall glucans [9,10]. The mechanism of *C. albicans* Spr1p is not fully understood, but it is expressed specifically during sporulation in *Saccharomyces cerevisiae* [11]. Exg2p is present during *C. albicans* cell wall regeneration. Xog1p is the major *exo*-1,3-β-glucanase associated with the *C. albicans* periplasmic cell wall. It is a non-glycosylated protein with a molecular mass of ∼45 kDa [12]. The cleavage of long chains of β-1,3 glucan causes softening of the cell wall. This enzyme is involved in cell wall remodeling of *Candida* that occurs during the swelling and germination of conidia or branching of hyphae [13]. The protein is involved in β-D-glucan metabolism and morphogenesis through its hydrolase and transglycosidase activities [14]. Targeting of this enzyme with sesamin can inhibit the morphogenetic switching of *Candida* spp. from non-virulent yeast to virulent hyphal form [14,15]. The structural complexity of cell wall components is essential for maintaining the physiology of *Candida* spp. A study has been conducted in which authors have reported the antifungal activity of edible sesame oil against *Candida albicans* by determining the effect of different concentrations of sesame oil on mycelial and yeast forms [15]. Targeting the function of the cell wall and its integrity is considered a promising approach to restrict the development of infection processes such as cell adhesion. Natural products are considered important medicinal candidates for the development of drugs because of their lower toxicity and side effects, low drug resistance, high bioavailability (oral intake), and better curative effects [16]. 

Accordingly, in this study, we have combined the screening of phytochemicals against different *Candida* species to identify innovative antifungal compounds. Using molecular docking and molecular dynamics (MD) simulation techniques, we have identified sesamin as a promising compound potentially able to interact with *exo*-1,3-β-glucanase. Sesamin has an immense therapeutic potential as an anticancer, anti-inflammatory, antiviral, antioxidant, and antimicrobial agent [17,18,19]. *Sesamum indicum* L. (Pedaliaceae) (sesame) is one of the most important crops in the world. Sesame oil contains large amounts of polyunsaturated fatty acids, proteins, carbohydrates, sesamin, sesamolin, sesamol, tocopherol, phytosterols, vitamins, and minerals [20,21]. Previous studies have investigated the natural product sesamol against different *Candida* spp. by reporting its effect on the calcineurin signaling pathway [22]. Some researchers have also reported the use of silver nanoparticles synthesized from sesame oil cake in anticancer and antimicrobial activities [23]. There was only a report of antifungal activity of sesamin against *C. albicans*, though the MIC value was reported to be very high [24]. 

This study, by combining computational techniques and biological experiments, provided a preliminary evaluation of sesamin as an antifungal agent against several *Candida* strains. The screening workflow is reported in Figure 1.

## 2. Results and Discussion

### 2.1. Computational Details

After the preparation of the library comprising 40 phytochemicals with favorable drug-like profiles (Appendix A), we performed a virtual ligand screening to determine whether the compounds behaved as *exo*-1,3-β-glucanase binders. The *exo*-1,3-β-glucanase (Exg) from *Candida* spp. is the product of EXG gene that encodes for a preproenzyme of 438 amino acids [25]. The mature protein comprises 400 amino acids and is involved in the metabolism of cell wall β-glucan of fungi by hydrolytic removal of a glucose residue from the non-reducing end of β-1,3-glucan and, to a much lesser extent, β-1,6-glucan [12]. Glucans are defined as structural components of fungal cell walls, usually in combination with other polymers providing rigidity and strength. Glucans are believed to have several cytoplasmic and extracellular functions. β-glucans probably act as carbon storage materials, which can be utilized under carbon limitation conditions, suggesting an important survival role [26]. The most widely accepted biological role of glucanase is limited to the hydrolysis of cell wall glucan during morphogenetic events. β-glucanase has been described to be associated with the *C. albicans* cell wall [14]. In *C. albicans*, *exo*-1,3-β-glucanase was found to be secreted and exported mainly during germ tube formation [27]. *Exo*-1,3-β-glucanase is responsible for most of the glucanase activity present in the *Candida* cell wall; thus, *exo*-1,3-β-glucanase can be considered as a possible drug target against *Candida* infections. β-glucanases are classified by the International Union of Biochemistry and Molecular Biology (IUBMB) (I.U.B, 1992) by the type of β-glucosidic linkages they hydrolyze and their hydrolytic action against specific substrates. Few crystallographic data are available for the study of this enzyme, although Cutfield and colleagues solved the structure of the *exo*-1,3-β-glucanase from *Candida albicans* in native and bound forms [14]. The active pocket of the enzyme is composed of eight highly conserved residues that define the active site of the glycosyl hydrolases family 5 (GH5), such as Arg92 (from the β2 strand), His135 (β3), Asn191, Glu192 (between β4 and α4), His253, Tyr255 (end of β6), Glu292 (β7), and Trp363 (end of β8) [14]. Recent active site labelling and mutagenesis experiments have identified the catalytic residues of the enzyme (Glu192 as proton donors and Glu292 as nucleophile) [28]. Accordingly, the active site includes two glutamate residues that act as a nucleophile and acid/base group [14]. *Exo*-1,3-β-glucanase enzyme is involved in cell wall β-glucan remodeling via its glucosyl hydrolase and transglycosylase activities. Exg represents transferase activity whereby an acceptor other than water can recover glucose from the glycosyl-enzyme intermediate [12]. These activities may be crucial for shaping the cell wall during morphogenesis. A double-displacement reaction involves the formation of a glycosyl-enzyme intermediate, and subsequent hydrolysis (or transglycosylation) proceeds via an oxocarbenium ion-like transition states [29]. The catalytic domain exhibits an extended binding cleft or groove supported by (β/α)_8_ barrel structure. Apart from these eight conserved residues around the catalytic center, the enzyme also contains aromatic residues that interact with β-1,3-glucan chain [29]. Entry of β-1,3-glucan chain to the pocket of *exo*-1,3-β-glucanase enzyme results in a change in the conformation of the terminal glucose residue from chair to twisted boat at the −1 position. However, the geometry of a pocket is not well suited for the cleavage of β-1,4-glycosidic linkage [14]. Notably, two glucose-binding sites have been identified. The first involves Glu27, while the second involves Phe144 and Phe258. The result of mutagenesis also depicts that the aromaticity afforded by phenylalanine residues provides a crucial property to clamp by altering k_cat_ and k_m_. It was found that the residue Phe144 and Phe258 pair controls both entry and release from the active site, which is both hydrolysis and transferase activity of the enzyme. An empty clamp of the enzyme allows rapid diffusion of hydrolyzed glucose, while an occupied clamp allows the acceptor to capture terminal glucose present in enzyme intermediate. Both Phe residues are highly conserved among all fungal *exo*-1,3-β-glucanases and are implicated in the specificity and catalytic efficiency of the enzyme. The results obtained from mutagenesis of the Phe–Phe gateway show that it not only maintains the aromatic nature of the entranceway, but also provides complete nonpolar and less bulky pairing substrate for the glycosidic bond cleavage at the non-reducing end [14,29]. With this information in mind, we considered the mentioned residues crucial to be targeted by potential ligands for obtaining strong binding to the enzyme as well as its potential inhibition. Accordingly, the visual inspection was conducted considering this kind of possible pattern of interaction involving Phe residues in hydrophobic contacts. The results of the virtual screening reported in Appendix A highlighted the most promising compounds that can potentially interact with the *exo*-1,3-β-glucanase enzyme.

The aim of molecular docking is to identify the potential binding mode of a ligand to a protein using a search algorithm. AutoDock 4.2 works on the principle of Lamarckian Genetic Algorithm (LGA) that helps in the study of protein–ligand conformation based on the lowest binding energies [30]. Following the estimation provided by molecular docking studies and after the visual inspection of the docked poses as previously mentioned, we ranked the compounds based on the computational score, as shown in Appendix A. The lowest computational score of the ligand with the selected receptor signifies that the ligand can satisfactorily interact with the considered enzyme. Each phytocompound was analyzed in terms of inhibition constant and the number of hydrogen bonds that can establish with the active site of the enzyme. Table 1 reports the name of phytocompounds, chemical structure, inhibition constant, and number of interacting residues involved in hydrogen bond formation. The more negative the binding energy, the greater the binding efficiency will be; thus, the considered ligand could effectively bind the selected drug target.

Based on the computational results, we identified sesamin (Table 1) as one of the most promising compounds due to its strong interaction within the selected binding site of the targeted enzyme and significant computational score. The various docking poses of sesamin as a hit compound with the selected fungal protein are described in Appendix A.

Table 2 shows the computational parameters along with the main interactions found within the active site of *exo*-1,3-β-glucanase, considering one of the most promising screened phytocompounds, sesamin. Inhibition constant is defined as the half maximum inhibition of an enzyme by a compound and is generally used to determine the potential of substrate or inhibitor in increasing/inhibiting the biological activity of enzymes. From the results, sesamin was found to have a predicted inhibition constant value of 1.11 nM. Notably, compounds with an inhibition constant value lower than 100 μM are defined as potential inhibitors, whereas compounds showing values greater than 100 μM are considered as non-potent inhibitors [31].

Considering the previously discussed mechanism of *exo*-1,3-β-glucanase enzyme, we report in Figure 2 the schematic representation of the possible interactions established by sesamin within the binding site of the enzyme. Considering that the residues Phe144 and Phe258 form a distinctive gateway approximately 10 Å up from the floor of the pocket defined by Trp363, this aromatic gateway acts as a clamp to control the entry of β-1,3-glucan substrate and the exit of free glucose product or, alternatively, glucosyl transfer to an acceptor [29]. Carbohydrate–protein interactions at the Phe–Phe clamp of *exo*-1,3-β-glucanase enzyme provide a major contribution in the transition state interaction energy. This sugar-binding site is crucial for substrate recognition. In the native *exo*-1,3-β-glucanase structure, the space between the phenylalanine rings is occupied by a series of largely disordered water molecules. The sugar moiety in ^4^C_1_ in chair conformation lies parallel between residues Phe144 and Phe258. This pocket is strongly targeted by sesamin (Figure 2 and Figure 3). Finally, the importance of the +1 subsite was observed through site-directed mutagenesis studies of the two phenylalanine residues involved in binding energies derived from kinetic studies [29].

Accordingly, the compound could behave as an *exo*-1,3-β-glucanase enzyme binder. In fact, as reported in Table 2, sesamin showed a relevant docking score with the selected fungal protein. Briefly, sesamin can target crucial residues in the binding site such as Phe144, Asn146, and Gly306 by H bonds, while it can establish strong hydrophobic interactions being able to form π–π stacking with Phe229 and Tyr255, while we detected a π–alkyl interaction with Phe258 (Figure 3). Moreover, sesamin was able to interact with the Phe–Phe clamp through π–alkyl and carbon–hydrogen bonds, as shown in Figure 3. Along with this, sesamin was also able to interact with residue Phe229, which makes the greatest contribution to the potential binding energy (Figure 3B). Accordingly, from the molecular docking studies, it has been observed that sesamin was able to target the active site of *exo*-1,3-β-glucanase of *Candida* spp by binding at the Phe144:Phe258 clamp and Phe229 residues, thus potentially blocking the entry of β-1,3-glucan chain and thus inhibiting the morphogenetic switching.

In order to validate the docking results and to assess the influence of binding of sesamin on the conformational dynamics of *exo*-1,3-β-glucanase, determining the stability of the complex and related timeline behavior, we conducted 100 ns MD simulation study on the *exo*-1,3-β-glucanase/sesamin complex. The obtained trajectory was evaluated using different standard simulation parameters, including the root-mean-square deviation (RMSD) assessment for each backbone atom and ligand and the root-mean-square fluctuation (RMSF) of each protein residue. The results of this study are illustrated in Figure 4A,B. The result analysis indicated that the system is quite stable (Figure 4A), with no relevant variations in structural stability observed upon binding of sesamin to the enzyme. Interestingly, sesamin maintained the interactions found from molecular docking studies, with no relevant movement within the binding site, preserving the interacting conformation (Figure 4A). The RMSF value represents the deviation between the protein atomic Cα co-ordinates and its average position throughout the MD run. The RMSF value is useful to determine the flexibility of specific protein backbone amino acids. As reported in Figure 4B, we detected a small fluctuation of the enzyme with the exclusion of some residues located at the C-terminus site.

To better describe the dynamic behavior of sesamin within the selected binding site, we analyzed the main contacts formed by the ligand within the active site, reporting the timeline analysis of MD simulation (Figure 5, panel A and B). The H bonds found by docking studies with Asp145 and Asn146 are maintained, although it became sporadically water-mediated. The hydrophobic contacts established by the ligand with Tyr29, Phe144, Phe229, Tyr255, and Phe258 were maintained during the simulation, providing strong hydrophobic network contacts, including several π–π stackings. Furthermore, during the MD simulation, we observed that the ligand established a novel significant H bond with the sidechain of Tyr255, contributing to stabilizing the retrieved binding mode. Overall, the computational outcome of the ligand within the selected binding site indicated that sesamin can behave as a potential ligand of the selected enzyme.

The computational output along a brief discussion for the other top-ranked compounds (pinoresinol, caflanone, and herbacetin) is described in the Appendix A file. In particular, in Appendix A, the in silico evaluation of pinoresinol within the *exo*-1,3-β-glucanase binding site is reported, in Appendix A, the result of the computational investigation regarding caflanone is shown, and, in Appendix A, the result of the computational studies regarding herbacetin is reported.

Furthermore, to provide further evidence on the possibility that sesamin and the other top-ranked compounds could act as an *exo*-1,3-β-glucanase binder, we calculated the ΔG_bind_ of phytocompounds to *exo*-1,3-β-glucanase using the whole trajectory obtained from MD studies. The output of this calculation is reported in Table 3. Based on the obtained data, we confirmed that sesamin could target the *exo*-1,3-β-glucanase enzyme, showing a satisfactory ΔG_bind_, and that the main contribution to the calculated ΔG_bind_ was represented from the hydrophobic interactions as suggested by the parameters ΔG_vdw_ and ΔG_Pack_. In addition, the top-ranked compounds showed satisfactory parameters in terms of binding energies.

The various properties of sesamin and other phytocompounds were analyzed using Swiss ADME server (Table 4). Through the mentioned tool, various parameters, including molecular weight, number of heavy atoms, rotatable bonds, H-bond donor, H-bond acceptors, aromatic heavy atoms, Csp^3^ fraction, molar refractivity, and total polar surface area (TPSA), were determined. The pharmacokinetic properties were determined for the selected compounds to establish the global drug-likeness. The study of gastrointestinal absorption passively by humans and the ability to cross the blood–brain barrier is important for drugs to be delivered at the place of requirement and is calculated through SwissADME. The metabolism of the compounds, considering the role of P-gp and five major isoforms of CYP450 (CYP1A2, CYP2C9, CYP2C19, CYP2D6, and CYP3A4), was assessed. Inhibition of any of these isoforms will result in drug accumulation due to the decrease in the clearance and metabolism of the drug. The SwissADME tool provides information in the form of ‘yes’ or ‘no’ showing whether the molecule has a high probability of being a substrate of P-gp or not along with a possible inhibitor of CYP450 isoforms. The drug-likeness was determined via five different filters. The filters include Lipinski (helps to determine preclinical development resulting in less failure of drug), Ghose (quantitatively estimates drug-likeness properties), Veber (helps to find out oral bioavailability of drug molecule), Egan (used to find out hydrogen bond donor), and Muegge (used for non-drug-like molecules). Moreover, the bioavailability score determines oral bioavailability of drug in a rat model, which lies in the permissible range, showing a value of 0.55 (Table 4).

The bioavailability radar plot of sesamin in Figure 6 demonstrates that the pink area is a suitable physicochemical space for oral bioavailability. The colored zone of the suitable physicochemical space shows the value of LIPO (lipophilicity); XLOGP3 (2.68), SIZE (molecular weight 354.35 g/mol), POLAR (polarity 55.38), INSOLU (insolubility, −3.93), and FLEX (flexibility, number of rotatable bonds 6) for the sesamin.

### 2.2. Biological Evaluation

#### 2.2.1. Drug Susceptibility Test Using Disc Diffusion Method

The drug susceptibility of human fungal pathogen to sesamin was determined using the disc diffusion method (Figure 7). The diameter of the zone of inhibition was calculated and experiments were conducted in triplicate. The zone of inhibition was found to be 5 mm and 5.2 mm for *Candida glabrata* at 16 μg/mL and 32 μg/mL concentration of sesamin, respectively. Meanwhile, with fluconazole (10 μg/mL), we observed a zone of inhibition of 4.8 mm for *Candida glabrata*.

#### 2.2.2. Minimum Inhibitory Concentration

The MIC and MFC values of sesamin against *Candida* spp. were found to be 16 and 32 µg/mL, respectively (Table 5). In particular, the in vitro antifungal susceptibility of sesamin was evaluated against *C. albicans* (ATCC 90028), *C. parapsilosis* (ATCC 22019), *C. krusei* (ATCC 6258), and *C. glabrata* (ATCC 15545).

Untreated control cells showed normal growth when compared with sesamin-treated cells.

#### 2.2.3. Spot Assay

Antifungal susceptibility was studied on solid media by spot assay (Figure 8, panel A and B) as one of the confirmatory methods. *Candida* cells were grown on YEPD agar for 48 h at 30 °C. The five spots in the horizontal direction represent the serial dilution of cell culture. There was a significant reduction in the growth of *Candida* cells, which was dependent on the sesamin concentration.

#### 2.2.4. Synergistic Study

Synergism is applied to drug combinations that can be defined as the interaction between two or more compounds that exhibit a greater effect than the additive sum of the effects of each drug when acting alone. The synergistic drug combination reduces the dose of single drug usage by increasing drug efficacy and reducing drug toxicity. Furthermore, the problem associated with the emergence of drug resistance can be reduced by multitarget strategy. The use of two or more antifungal drugs to combat severe invasive fungal infections has been accepted as a better alternative option in clinics for a long time. The first reported implementation of synergistic therapy for invasive candidiasis is flucytosine and amphotericin B [32]. In the present work, synergistic experiments with sesamin were performed employing fluconazole against different *Candida* spp. The calculated FICI value of fluconazole (main marketed antifungal drug in India) along with sesamin was lower than 0.5, suggesting a synergistic mode of interaction between the two drugs (Table 6).

## 3. Materials and Methods

### 3.1. Computational Details

#### 3.1.1. Phytochemical Library Preparation

Suitable phytochemicals were selected from PubChem, Google Scholar, and ScienceDirect, using keywords such as ‘phytochemicals’ and ‘anti-fungal activity’. Following this searching method, we generated a small library containing 40 natural phytocompounds belonging to different classes, such as phenols, flavonoids, tannins, terpenoids, alkaloids, quinones, coumarins, and lignans, with favorable Lipinski’s rule (Appendix A) [33].

#### 3.1.2. Molecular Docking

The 3D structure of the fungal protein *exo*-1,3-β-glucanase (PDB ID 4M80) was retrieved from the Protein Database Bank [34]. The 3D structures of all compounds in the selected library were converted to pdbqt files using MGL tools. Molecular docking was performed using AutoDock 4.2 software via AutoDock Tools-1.5.7 [30]. Briefly, the protein PDB file was opened in AutoDock Tools. The water molecules and any bound ligand were deleted, followed by addition of Kollmann charges to the protein molecule. The grid parameter was selected to cover the active site of protein for analysis. In this study, the grid spacing was set to 0.375 Å (default) and center grid box values were set to x = 4.075, y = 64.791, and z = 9.706, respectively. The number of grid points along x, y, and z dimensions was set as 48 × 62 × 46. After the successful completion of Autogrid, the Genetic Algorithm (GA) was set to default as follows: (1) number of GA runs, 50; (2) population size, 100; (3) number of energy evaluations, 2.5 million (2.0 Å clustered tolerance); and (4) number of generations, 27000. After that, the docking complex was retrieved from AutoDock Tools in pdbqt file format. Interactions between ligands and the selected protein were calculated on the basis of the docking score (i.e., binding energy, kcal/mol) and inhibition constant (nM), along with the interacting protein residues. The Computed Atlas for Surface Topography of Proteins (CASTp) [35] and Biovia Discovery Studio (Discovery Studio, version 4.5, San Diego: Dassault Systèmes, Biovia, 2019) were used to determine the amino acids present in the active site of the protein.

#### 3.1.3. Molecular Dynamics (MD)

MD simulations were performed using Desmond 5.6 academic version (Desmond Molecular Dynamics System, version 5.6, D. E. Shaw Research, New York, NY, 2018. Maestro-Desmond Interoperability Tools, Schrödinger, New York, NY, USA, 2018) as previously described by us [36,37]. The complex retrieved from the molecular docking investigation was solvated into an orthorhombic box filled with water (TIP3P model) using Desmond system builder. MD simulation was performed adopting OPLS as the force field, using monovalent ions (Na^+^ and Cl^−)^, physiological concentration of 0.15 M, at constant temperature (310 K) and pressure (1.01325 bar) with the ensemble class NPT (constant number of particles, pressure, and temperature). A RESPA integrator was applied to integrate the equations of motion (inner time step of 2.0 fs). Nose–Hoover thermostats were used to keep a constant simulation temperature, and the Martyna–Tobias–Klein method was applied to control the pressure. Long-range electrostatic interactions were calculated by particle-mesh Ewald method (PME) with 9.0 Å for van der Waals and short-range electrostatic interactions. The equilibration of the system was performed employing the default protocol provided in Desmond. Consequently, one individual trajectory of 100 ns was calculated. MD simulation experiments were repeated twice to provide more reliable output. The trajectory files were analyzed by Simulation Event implemented in the Desmond package, which was also used to generate all plots regarding the MD simulation experiment presented in this study. Furthermore, we calculated the binding energy (ΔG_bind_) of sesamin within the target protein from the trajectory obtained by Desmond, using the thermal MM-GBSA script available in Desmond (*thermal_mmgbsa.py*). This tool uses the Desmond MD trajectory, splitting it into individual frame snapshots, and runs each one through MM-GBSA analysis. During the MM-GBSA calculation, 1000 snapshots from 100 ns MD simulation were used as input to compute the average binding free energy. The evaluated ΔG_bind_ is reported as the average value in Section 2 along with the energy components used in the calculation.

#### 3.1.4. ADMET/Pharmacokinetics Predictions and Bioactive Evaluation

The ADMET properties of the most promising phytocompound sesamin were evaluated using the Swiss ADME tool [38]. The interaction of ligand with enzyme or protein cannot ensure it as an effective drug. Drug-likeness studies help us to measure the probability of a molecule to act as an oral drug in terms of its bioavailability. The knowledge of various properties is essential for the process of drug design and the presence of unacceptable ADMET properties leads to failure of drug in clinical trials. ADMET gives accurate properties of the drug-likely nature of the phytocompound and provides numerical values of the parameters such as blood–brain barrier permeation, gastrointestinal absorption, and provides information about toxicity tests such as the Ames test, mouse carcinogenicity, and *h*ERG inhibition.

### 3.2. Biological Evaluation of Sesamin as an Antifungal Agent

#### 3.2.1. Growth Media and Strains Used

The strains of *Candida* used in this study were *C. krusei* (ATCC 6258), *C. albicans* (ATCC 90028), *C. parapsilosis* (ATCC 22019), and *C. glabrata* (ATCC 15545). All *Candida* strains were stored in 30% (*v*/*v*) glycerol stock at −80 °C. The cells were freshly revived on Sabouraud’s dextrose agar (SDA) plate and transferred to YEPD broth. The cells were grown at 30 °C on agar plate before each study to ensure the revival of the strains. All strains of *Candida* were cultured in YEPD broth with the composition of yeast extract 1% (*w*/*v*), peptone 2% (*w*/*v*), and dextrose 2% (*w*/*v*). For agar plates, 2% (*w*/*v*) agar (HiMedia, Mumbai, India) was added to the media. Sesamin was commercially purchased from Combi Blocks (Purity 98%).

#### 3.2.2. Inoculum Preparation

The suspensions were prepared from fresh *Candida* cultures, plated on agar plates, and incubated at 35 °C for 24–48 h. After incubation, we transferred roughly 4–5 yeast colonies (with a sterile loop) to a test tube containing 3 mL of 0.85% saline solution. The resulting suspensions were vortexed for 15 s. The final concentration obtained was 1–5 × 10^6^ colonies.

#### 3.2.3. Disc Diffusion Assay

The disc diffusion assay was performed as described [39]. The drugs were spotted in a volume of 10 µL at the desired concentration, and the diameters of the respective zones of inhibition were measured after incubation of the plates for 48 h at 30 °C.

#### 3.2.4. Spot Assay

For the spot assay, 5 µL of fivefold serial dilutions of each yeast culture (each with cells suspended in normal saline to an OD_600_ nm of 0.1) was spotted onto YEPD plates in the absence (control) and presence of the drugs. The growth difference was measured after incubation at 30 °C for 48 h [40].

#### 3.2.5. Determination of the Minimum Inhibitory Concentration (MIC) and Minimum Fungicidal Concentration (MFC)

MIC was determined by the broth dilution method as described in method M27-S4 from the Clinical and Laboratory Standards Institute (CLSI), formerly NCCLS [41]. Briefly, 100 µL of media was placed in each well of the 96-well plate following the addition of the drug with the remaining media and was then serially diluted. A total of 100 µL of cell suspension (in normal saline to an OD_600_ 0.1) was added to each well of the plate. In parallel, controls were established for yeast viability and susceptibility to fluconazole and amphotericin B. The plates were incubated at 35 °C for 24–48 h. After the incubation period, the presence or absence of growth was observed visually. MIC was defined as the lowest concentration that produced a visible inhibition of yeast growth. MFC was defined as the lowest concentration of the test compound that completely inhibited the growth of yeast or caused less than three colony forming units (CFUs) to occur, resulting in fungicidal activity.

#### 3.2.6. Synergistic Study/Checkerboard Microdilution Assay

The antifungal activity of sesamin in combination with a standard antifungal drug fluconazole was evaluated by a synergistic study. A total of 50 µL of sesamin (64 to 0.5 µg/mL) and 50 µL of fluconazole (64 to 0.125 µg/mL) were added to a microtiter plate. Each well was inoculated with 100 µL of fungal suspension to make up a final volume of 200 µL. The obtained checkerboard plates were incubated overnight at 37 °C. The fractional inhibitory concentration index (FICI) was calculated using the following equation [42]:$$\mathrm{FIC}\;\mathrm{Index}=\frac{\mathrm{MIC}\;\mathrm{of}\;\mathrm{Sesamin}\;\mathrm{in}\;\mathrm{combination}}{\mathrm{MIC}\;\mathrm{of}\;\mathrm{Sesamin}\;\mathrm{alone}}+\frac{\mathrm{MIC}\;\mathrm{of}\;\mathrm{Fluconazole}\;\mathrm{in}\;\mathrm{combination}}{\mathrm{MIC}\;\mathrm{of}\;\mathrm{Fluconazole}\;\mathrm{alone}}$$

The ‘synergy and antagonism’ were defined by FICI ≤ 0.5 and > 4, respectively. Meanwhile, ‘partial synergistic’ was defined by 0.5 < FICI < 1, whereas ‘indifference’ was defined by 1 < FICI ≤ 4.

## 4. Conclusions

The fungal cell wall is mainly composed of chitin, glucans, mannans, and glycoproteins required for adhesion. It also acts as a protective barrier. Targeting cell wall components and their enzymes can be attractive drug targets for developing novel antifungal agents. In this work, we have developed an integrated screening platform based on in silico methods and comprehensive biological evaluation to identify novel antifungal agents. We used this platform to select natural products that are able to act as antifungal agents by targeting *exo*-1,3-β-glucanase enzyme present in *Candida* spp. The *exo*-1,3-β-glucanase enzyme is involved in β-D-glucan metabolism and eventual fungal morphogenesis through its hydrolase and transglycosidase activities. The yeast hyphae morphological transition is defined as a highly virulent feature involved in fungal pathogenesis of host tissues. Following our computational protocol based on molecular docking, MD, and evaluation of drug-like profiles of the compounds, we identified sesamin as a potential *exo*-1,3-β-glucanase binder endowed with a possible antifungal profile. We further confirmed the in silico findings by conducting a series of in vitro tests to assess the activity of sesamin against different *Candida* species. Sesamin showed an interesting antifungal profile with an MIC value of 16 μg/mL. A previous study by Agbo et al. has determined a much higher MIC value of sesamin against *Candida albicans* (256 μg/mL) [24], which makes the present work extremely relevant. We also performed synergistic studies of sesamin with reference drug fluconazole, obtaining interesting results. Accordingly, the present study highlights the natural product sesamin as an interesting scaffold for developing novel therapeutic agents against fungal infections due to its low toxicity, easy availability, and significant antifungal activity.

## Figures and Tables

**Figure 1 molecules-28-04658-f001:**
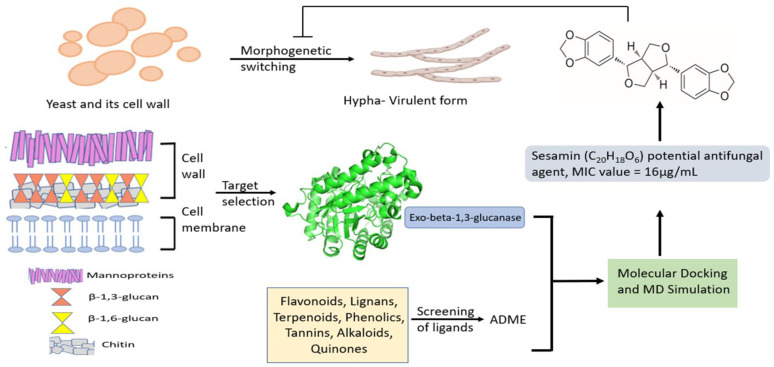
Screening workflow to identify novel antifungal agents.

**Figure 2 molecules-28-04658-f002:**
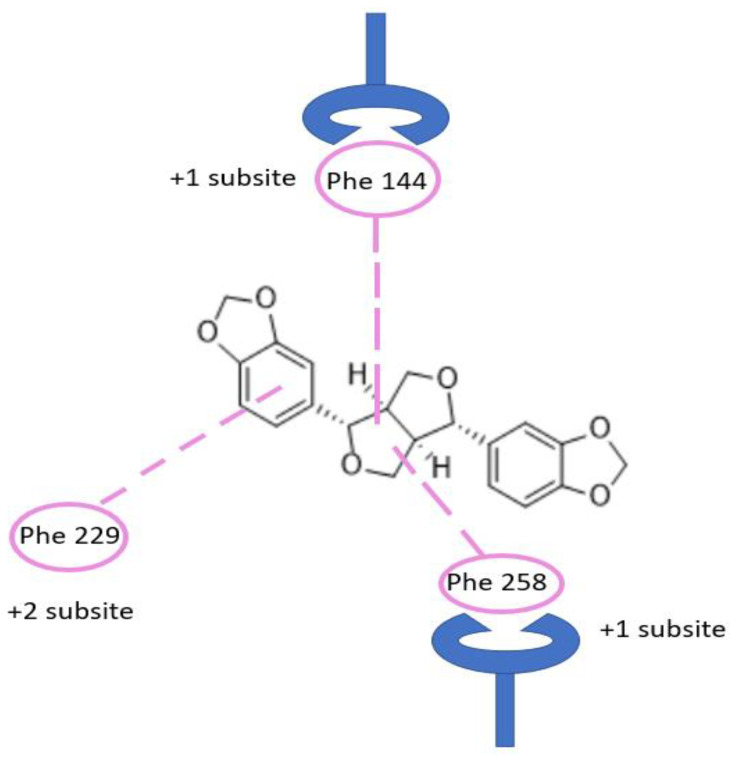
Binding of sesamin in the active site pocket showing Phe–Phe clamp and Phe229. Carbohydrate binding of β-1,3-glucan chain occurred at this −1, +1 subsite, and +2 subsite, while glycosidic bond cleavage occurs at −1 and +1 subsite.

**Figure 3 molecules-28-04658-f003:**
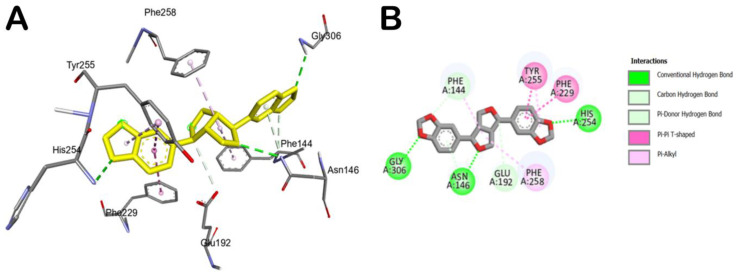
Main interactions of sesamin within the *exo*-1,3-β-glucanase binding site. (**A**) 3D model of interaction. (**B**) 2D schematic representation of the main interactions established by sesamin within the selected binding site of *exo*-1,3-β-glucanase.

**Figure 4 molecules-28-04658-f004:**
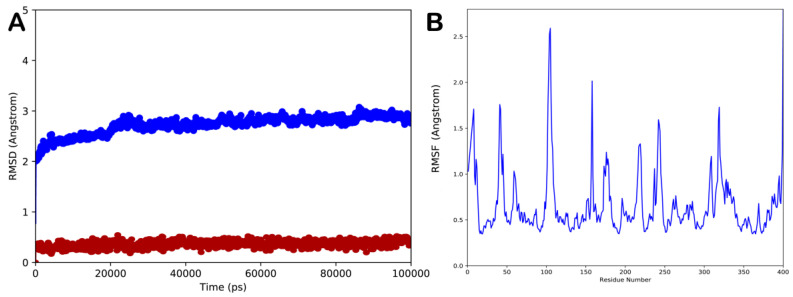
(**A**) Root-mean-square deviation (RMSD) regarding the protein/ligand complex (blue line for the protein and red line for the ligand); (**B**) root-mean-square fluctuation (RMSF) of all residues of the protein. Pictures were created by Simulation Event Analysis available in Desmond.

**Figure 5 molecules-28-04658-f005:**
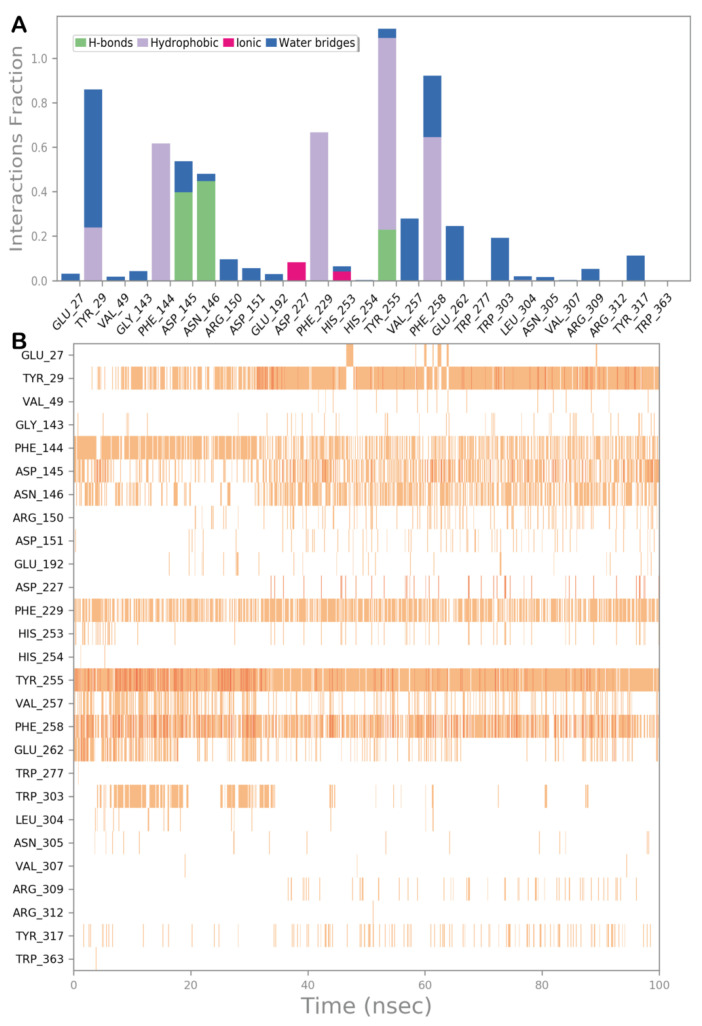
(**A**) Protein–ligand interactions monitored throughout the MD simulation. The interactions can be grouped into four types: H bonds (green), hydrophobic (grey), ionic (magenta), and water bridges (blue). (**B**) The diagram illustrates a timeline description of the main interactions. A darker hue of orange indicates that some residues make many distinct contacts with the ligand.

**Figure 6 molecules-28-04658-f006:**
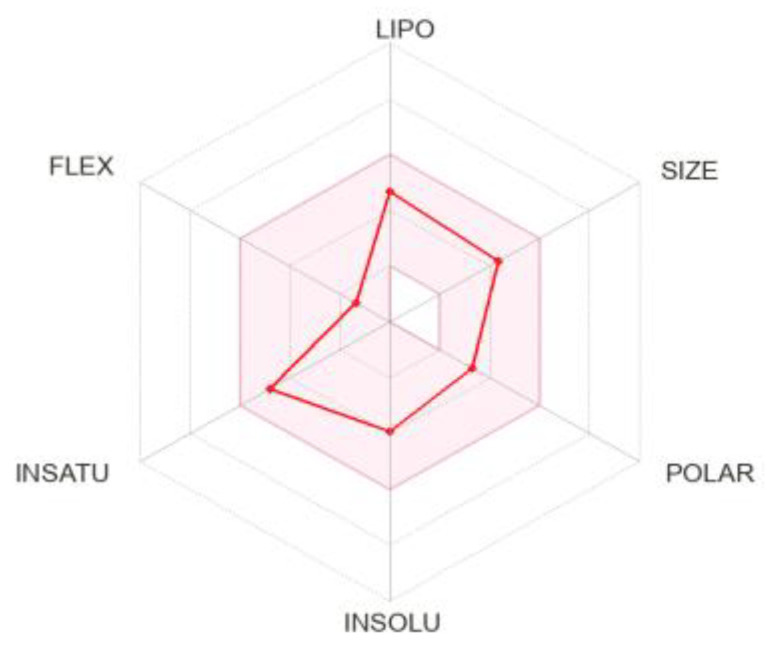
Bioavailability radar plot of sesamin; the pink area is suitable physicochemical space for oral bioavailability LIPO (lipophilicity): −0.7 < XLOGP3 < +5.0, SIZE: 150 g/mol < MW < 500 g/mol, POLAR (polarity): 20 < TPSA < 130, INSOLU (insolubility): −6 < Log S (ESOL) < 0, INSATU (insaturation): 0.25 < Fraction Csp^3^ < 1, FLEX (flexibility): 0 < Num. rotatable bonds < 9.

**Figure 7 molecules-28-04658-f007:**
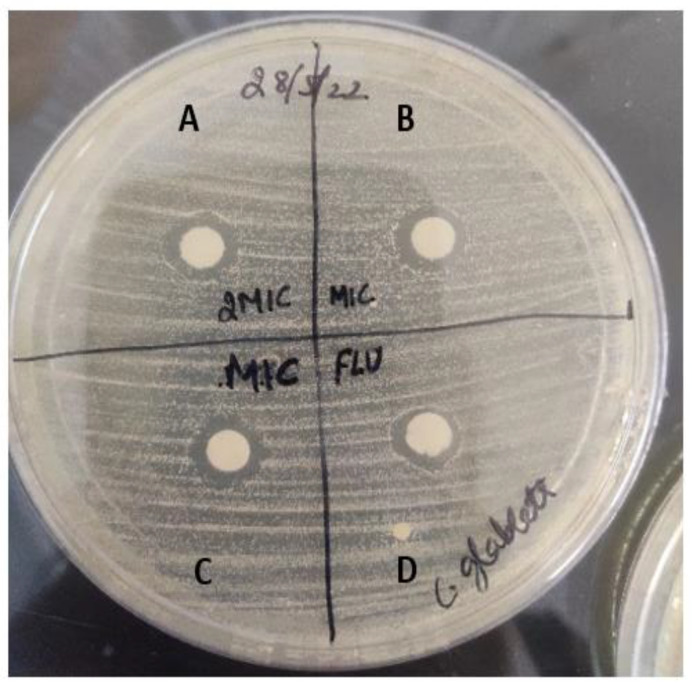
Zone of inhibition formed by drug molecule against human fungal pathogen, *Candida glabrata*. A represents zone of inhibition at 32 μg/mL of sesamin concentration, B and C represent the zone of inhibition at 16 μg/mL of sesamin concentration, and D represents positive control fluconazole drug (10 μg/mL).

**Figure 8 molecules-28-04658-f008:**
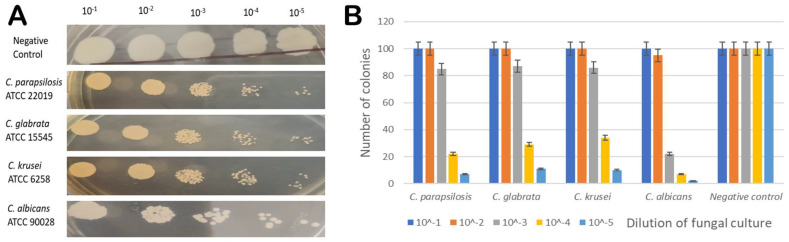
(**A**) Spot assay profile of *C. parapsilosis* (ATCC 22019), *C. glabrata* (ATCC 15545), *C. krusei* (ATCC 6258), *C. albicans* (ATCC 90028) cells in the presence of MIC concentration of sesamin, and untreated *C. parapsilosis* (ATCC 22019) as negative control. (**B**) Graphical representation of spot assay results in the presence of MIC concentration of sesamin (at 16 μg/mL) (*x* axis—*Candida* spp. (dilution) and *y* axis—number of colonies).

**Table 1 molecules-28-04658-t001:** Comparison of docking results of the four potential novel inhibitors of *exo*-1,3-β-glucanase.

Name of Compound	Chemical Structure	Binding Energy (kcal/mol)	Inhibition Constant (nM)	Interacting Residues Involved in H Bonds
Sesamin (Lignan)	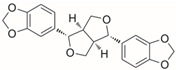	−12.21	1.11	His254, Asn146, Gly306
Pinoresinol (Lignan)	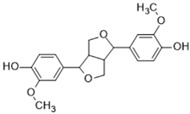	−11.50	3.71	Asp145, Asn146, His254
Caflanone (Flavonoid)	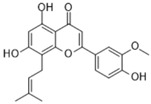	−11.48	3.80	Asp145, Tyr255
Herbacetin (Flavonoid)	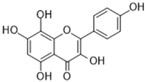	−11.03	8.21	Glu27, Glu192, Ser292, Leu304

**Table 2 molecules-28-04658-t002:** Docking results of novel inhibitor (sesamin) against *exo*-1,3-β-glucanase (PDB ID: 4M80).

Compound	Pocket Grid Dimension for Molecular Docking Studies	Docking Score (kcal/mol)	Inhibition Constant (nM)	Residue Involved in Various Interactions
Sesamin	Grid-box center coordinates Center-x = 4.075 Center-y = 64.791 Center-z = 9.706 Grid-box Size size-x = 48 size-y = 62 size z = 46	−12.21	1.11	Phe144, Asp145, Asn146, Glu192, Phe229, Tyr255, Phe258, Gly306

**Table 3 molecules-28-04658-t003:** Predicted free binding energies (ΔG_bind_) from the MM-GBSA calculation and the energy components of the top-ranked compounds in complex with *exo*-1,3-β-glucanase.

Entry	ΔG_vdw_ ^a^ (kcal/mol)	ΔG_coul_ ^b^ (kcal/mol)	ΔG_Hbond_ ^c^ (kcal/mol)	ΔG_Lipo_ ^d^ (kcal/mol)	ΔG_Pack_ ^e^ (kcal/mol)	ΔG_SolGB_ ^f^ (kcal/mol)	ΔG_bind_ ^g^ (kcal/mol)
Sesamin	−45.74	−14.29	−1.14	−29.53	−4.91	26.31	−61.67
Pinoresinol	−43.91	−12.11	−1.07	−26.82	−2.11	24.67	−59.31
Caflanone	−35.17	−9.23	−1.04	−27.14	−0.73	19.16	−53.22
Herbacetin	−34.28	−13.48	−3.22	−28.79	−4.31	20.37	−58.87

^a^ Contribution of van der Waals interaction energy to the binding free energy; ^b^ contribution of Coulomb energy to the binding free energy; ^c^ hydrogen-bonding contribution to the binding free energy; ^d^ contribution of the lipophilic energy to the binding free energy; ^e^ π−π packing energy contribution to the binding free energy; ^f^ generalized Born electrostatic solvation energy contribution to the binding free energy; ^g^ total binding free energy.

**Table 4 molecules-28-04658-t004:** Drug-likeness and ADME profile of top-ranked phytocompounds.

	Molecular Descriptor	Pinoresinol (Lignan)	Caflanone (Flavonoid)	Herbacetin (Flavonoid)	Sesamin (Lignan)	Fluconazole (Commercial Drug)
**Physicochemical parameter**						
	Molecular formula	C_20_H_22_O_6_	C_21_H_20_O_6_	C_15_H_10_O_7_	C_20_H_18_O_6_	C_13_H_12_F_2_N_6_O
	Number of rotatable bonds	4	4	1		5
	Molar refractivity	94.90	104.20	78.03	90.00	70.71
	Fraction Csp^3^	0.40	0.19	0.00	0.40	0.23
	Number of heavy atoms	26	27	22	26	22
	Number of aromatic heavy atoms	12	16	16	12	16
	TPSA	77.38 Å^2^	100.13 Å^2^	131.36 Å^2^	55.38 Å^2^	81.65 Å^2^
**Lipophilicity**						
	Log P _o/w_ (iLOGP)	2.67	3.29	1.50	3.46	0.41
	Log P _o/w_ (XLOGP3)	2.28	5.03	2.17	2.68	0.35
	Log P _o/w_ (WLOGP)	2.54	4.09	1.99	2.57	1.47
	Log P _o/w_ (MLOGP)	1.17	1.31	−0.56	1.98	1.47
	Log P _o/w_ (SILICOS-IT)	2.66	4.28	1.54	3.25	0.71
	Consensus Log P _o/w_	2.26	3.60	1.33	2.79	0.88
**Pharmacokinetics**						
	GI-absorption	High	High	High	High	High
	BBB permeant	Yes	No	No	Yes	No
	P-gp substrate	Yes	No	No	No	Yes
	CYP1A2 inhibitor	No	No	Yes	No	No
	CYP2C19 inhibitor	No	No	No	Yes	Yes
	CYP2C9 inhibitor	No	Yes	No	No	No
	CYP2D6 inhibitor	Yes	No	Yes	Yes	No
	CYP3A4 inhibitor	Yes	No	Yes	Yes	No
	Log K_p_ (skin permeation)	−6.87 cm/s	−4.98 cm/s	−6.60 cm/s	−6.56 cm/s	−7.92 cm/s
**Druglikeness**						
	Lipinski	Yes; 0 violation	Yes; 0 violation	Yes; 0 violation	Yes; 0 violation	Yes: 0 violation
	Ghose	Yes	Yes	Yes	Yes	Yes
	Veber	Yes	Yes	Yes	Yes	Yes
	Egan	Yes	Yes	Yes	Yes	Yes
	Muegge	Yes	No; 1 violation: XLOGP3>5	Yes	Yes	Yes
	Bioavailability score	0.55	0.55	0.55	0.55	0.55

**Table 5 molecules-28-04658-t005:** In vitro antifungal susceptibility of sesamin (values in μg/mL).

Name of Strains	Sesamin		Fluconazole		Amphotericin B	
	MIC	MFC	MIC	MFC	MIC	MFC
*C. albicans* (ATCC 90028)	16	32	2	4	1	2
*C. glabrata* (ATCC 15545)	16	32	1	2	0.5	2
*C. krusei* (ATCC 6258)	16	32	16	32	1	2
*C. parapsilosis* (ATCC 22019)	16	32	1	2	0.5	2

**Table 6 molecules-28-04658-t006:** Synergistic study considering the identified compound sesamin with marketed drug fluconazole.

Name of Strain	MIC (µg/mL)					
	Alone Sesamin	Alone Fluconazole	Combination Sesamin+ Fluconazole		FICI	Interaction Mode
*C. albicans* (ATCC 90028)	16	2	0.125	0.125	0.07	Synergistic
*C. parapsilosis* (ATCC 22019)	16	2	0.125	0.125	0.07	Synergistic
*C. krusei* (ATCC 6258)	16	16	0.5	0.125	0.03	Synergistic
*C. glabrata* (ATCC 15545)	16	1	0.125	0.125	0.13	Synergistic

## Data Availability

Not applicable.

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
