# Peer review of "Identification of Sesamin from Sesamum indicum as a Potent Antifungal Agent Using an Integrated in Silico and Biological Screening Platform"

_molecules, 2023, doi:10.3390/molecules28124658_

Round 1

Reviewer 1 Report (Previous Reviewer 1)

The manuscript was corrected according to the suggestions.

Author Response

The manuscript was corrected according to the suggestions.

Authors: we thank the reviewers for appreciating the revisions done.

Reviewer 2 Report (New Reviewer)

In this manuscript, the authors employed docking and molecular dynamic simulations to identify a highly effective inhibitor of exo-1,3,-b-glucanase. Additionally, they evaluated its antifungal activity. The manuscript emphasizes the potential applications of their screening protocol, which encompasses both computational and biological screening, in the realm of antifungal drug discovery.

Here, I have several questions and suggestions:

1, It is recommended that the authors consider analyzing multiple docking poses of the hit compound instead of solely focusing on the one with the lowest docking score. Exploring various poses may provide valuable insights and a more comprehensive understanding of the binding interactions after molecular dynamic simulations. Relying solely on the docking score might not be entirely reliable for selecting poses.

2, Do the authors have analogs of sesamin? Conducting a preliminary Structure-Activity Relationship (SAR) analysis using analogs could help confirm the docking pose and provide additional evidence for the binding mode.

3, The inclusion of positive and negative controls in the computational prediction of the pharmacokinetic (pk) and ADMET (Absorption, Distribution, Metabolism, Excretion, and Toxicity) properties is suggested. This addition would enhance the reliability of the computational predictions.

4, To improve the clarity and readability of the manuscript, it is advisable for the authors to simplify Table 1. Consider moving some of the detailed information to the Supporting Information section, while retaining the essential and most informative data in the main table.

5, Transforming Figure 8 into a statistical analysis figure, rather than solely presenting pictures, would be beneficial. This could include appropriate statistical analysis methods, such as error bars or box plots, to provide a more comprehensive and quantitative representation of the data.

The language part is good.

Author Response

In this manuscript, the authors employed docking and molecular dynamic simulations to identify a highly effective inhibitor of exo-1,3,-b-glucanase. Additionally, they evaluated its antifungal activity. The manuscript emphasizes the potential applications of their screening protocol, which encompasses both computational and biological screening, in the realm of antifungal drug discovery.

Here, I have several questions and suggestions:

1, It is recommended that the authors consider analyzing multiple docking poses of the hit compound instead of solely focusing on the one with the lowest docking score. Exploring various poses may provide valuable insights and a more comprehensive understanding of the binding interactions after molecular dynamic simulations. Relying solely on the docking score might not be entirely reliable for selecting poses.

Authors: we thank the reviewer for the valuable comments that allowed us to improve the quality of the paper. Considering the comments, we have added these results depicting multiple docking poses of hit compound in supplementary file (Table S2). Accordingly, the binding poses showed a limited difference considering the docking scores. Furthermore, the selected binding pose for MD simulation studies highlighted a relevant stability concerning the binding mode, providing good reliability of the proposed results.

2, Do the authors have analogs of sesamin? Conducting a preliminary Structure-Activity Relationship (SAR) analysis using analogs could help confirm the docking pose and provide additional evidence for the binding mode.

Authors: we thank the reviewer fort he suggestion. At the moment we don’t have analogs of Sesamin, but this will be part of our next study. We would like to highlight that in the screening step we identified also pinoresinol that is defined as the precursor of sesamin and due to the high structural similarity with sesamin and based on the computational outcome, we expected some antifungal activity also for this compound. Anyway, the step for developing reliable SAR study will start from this work to find further analogs of sesamin for conducting an inclusive Sar analysis.

3, The inclusion of positive and negative controls in the computational prediction of the pharmacokinetic (pk) and ADMET (Absorption, Distribution, Metabolism, Excretion, and Toxicity) properties is suggested. This addition would enhance the reliability of the computational predictions.

Authors: According to the reviewer comment, we have added the ADME profile of Fluconazole (Commercial antifungal drug) in Table 4.

4, To improve the clarity and readability of the manuscript, it is advisable for the authors to simplify Table 1. Consider moving some of the detailed information to the Supporting Information section, while retaining the essential and most informative data in the main table.

Authors: According to the reviewer comment, we moved the Table 1 in the supporting information (now Table S1), maintaining Table 2 (now Table 1) with the docking results for the four top-ranked compounds.

5, Transforming Figure 8 into a statistical analysis figure, rather than solely presenting pictures, would be beneficial. This could include appropriate statistical analysis methods, such as error bars or box plots, to provide a more comprehensive and quantitative representation of the data.

Authors: According to the reviewer comment, we revised Figure 8 adding the graphical representation of spot assay results.

This manuscript is a resubmission of an earlier submission. The following is a list of the peer review reports and author responses from that submission.

Round 1

Reviewer 1 Report

The authors present a mixed strategy work (computational-experimental) to report sesamin as a potent 2 antifungal agent. In general, the proposal will be of interest to those working in fungicidal drug design.  However, there are some points that deserve to be analyzed.

1.- Why compounds with a better binding score than fluconazole did not were studied? or at least those with the closest binding score to sesamine, such as caflanone and pinoresinol? It is correct that sesamin works as a fungicidal agent acting at the level of cell wall, according to the evidence, but it would be interesting to analyze if at least the other two compounds work in the same way, because their effect on the enzyme was not assessed and the mechanism of sesamine could be a coincidence and not as a consequence of the computational protocol. Therefore, to give support it is necessary to study minimum two more compounds.

2.- In the same context, why the effect of the compounds was not assessed in the enzyme?

3.- Nothing is mentioned or discussed about data showed in figure 8, what about disc diameter in the different conditions? please include values.

4.- Why MIC values showed in Table 6 are different to those showed in Table 4 in equivalent conditions (without sorbitol)?

5.- Why the MIC values in Table 6 are the same in 2 or 7 days?

6.- In methodology, molecular docking section, is mentioned the determination of an “inhibition constant (μM)” but nothing is given in the manuscript, what about this?

7.- If molecular dynamics was performed, why do not estimate the “binding energy” from these data?

8.- In general, additional to the experiments mentioned in point 1, a deeper discussion of the results is necessary.  

Additional minor details:

1.- The word “predicted” must be added before …pharmacological profile, in line 28.

2.- Information in lines 71-75 and 80-83 must be referenced.

3.- Figures 3, 4, and 6 need to be divided in subsections (A and B).

4.- Information in figure 7 is equivalent to that showed in table 3, I suggest delete this figure.

5.- The title of table 4 is too long, it contains information provided in the text or inside the table, maybe “In vitro antifungal susceptibility of sesamin” is enough.

6.- Figure legend in figure 8 needs to include the name of the microorganism.

7.- In table 1, the value calculated is a “binding score” not a “binding energy” as is correctly stated in lines 111-112.

Author Response

The authors present a mixed strategy work (computational-experimental) to report sesamin as a potent 2 antifungal agent. In general, the proposal will be of interest to those working in fungicidal drug design.  However, there are some points that deserve to be analyzed.

1.- Why compounds with a better binding score than fluconazole did not were studied? or at least those with the closest binding score to sesamine, such as caflanone and pinoresinol? It is correct that sesamin works as a fungicidal agent acting at the level of cell wall, according to the evidence, but it would be interesting to analyze if at least the other two compounds work in the same way, because their effect on the enzyme was not assessed and the mechanism of sesamine could be a coincidence and not as a consequence of the computational protocol. Therefore, to give support it is necessary to study minimum two more compounds.

Authors: authors thank the reviewer for the suggestions. We would like to highlight that Pinoresinol is defined as the precursor of sesamin, while caflanone is the part of another study to explore its antifungal nature, currently ongoing in our lab.

2.- In the same context, why the effect of the compounds was not assessed in the enzyme?

Authors: As argued from the referee, this step could be important to better comprehend the exact mechanism of action of sesamin as antifungal agent. However, for evaluating the capability of sesamin to target the proposed enzyme a commercial kit of enzyme is not available. In order to perform this experiment in our lab, we are arranging the components to perform this test, but long time is necessary. We have tried to move the deadline for the submission of the revised article but in an additional month, we are not able to have an efficient test to evaluate compound ready for the application, yet. Obviously, we are doing our best to further characterize sesamin and its predicted target. Accordingly, in this part of our research, we are concluding our results of Sesamin on the basis of in silico work and its antifungal activity.

3.- Nothing is mentioned or discussed about data showed in figure 8, what about disc diameter in the different conditions? please include values.

Authors: thank you for your suggestion. Details about the data showed in Fig 8 have been added in the main text, accordingly.

4.- Why MIC values showed in Table 6 are different to those showed in Table 4 in equivalent conditions (without sorbitol)?

Authors: thank you for your suggestion. In order to confirm the data we reperformed the test. Unfortunately, we did not obtain reproducible results, although, sesamin demonstrated significant activity in this test. Considering the issue, we preferred to delete the study with sorbitol, in order to better investigate the sesamin using this test in the future.

5.- Why the MIC values in Table 6 are the same in 2 or 7 days?

Authors: as previously mentioned the data enclosed in table 6 has been removed.

6.- In methodology, molecular docking section, is mentioned the determination of an “inhibition constant (μM)” but nothing is given in the manuscript, what about this?

Authors: we detailed the inhibition constant (μM) in the materials and methods. However, it is a parameter calculated by autodock along with the binding affinity.

7.- If molecular dynamics was performed, why do not estimate the “binding energy” from these data?

Authors: thank you for the suggestion. According to the referee comments, we calculated the binding energy from the trajectory obtained by Desmond, using the script thermal_mmgbsa.py. The thermal MM-GBSA script available in Desmond (thermal_mmgbsa.py) was used to evaluate the ΔGbind for the selected complexes. This tool used the Desmond MD trajectory, splitting it into individual frame snapshots, and runs each one through MM-GBSA analysis. During the MM-GBSA calculation, 1000 snapshots from the 100 ns MD simulation were used as input to compute the average binding free energy. The evaluated ΔGbind are reported as average values in the Results and Discussion section along with the energy components used in the calculation.

8.- In general, additional to the experiments mentioned in point 1, a deeper discussion of the results is necessary.  

Authors: we thank the referee for the suggestion. We revised the discussion on the obtained results to give a more information to the reader.

Additional minor details:

1.- The word “predicted” must be added before …pharmacological profile, in line 28.

Authors: predicted was added according to the referee suggestion.

2.- Information in lines 71-75 and 80-83 must be referenced.

Authors: references have been added in the highlighted lines

3.- Figures 3, 4, and 6 need to be divided in subsections (A and B).

Authors: the figures have been divided in subsections.

4.- Information in figure 7 is equivalent to that showed in table 3, I suggest delete this figure.

Authors: we prefer to maintain figure 7 since it is a graphical representation of the ADME properties considering other parameters with respect to those reported in table 3.

5.- The title of table 4 is too long, it contains information provided in the text or inside the table, maybe “In vitro antifungal susceptibility of sesamin” is enough.

Authors: the title was revised accordingly, inserting the details in the main text.

6.- Figure legend in figure 8 needs to include the name of the microorganism.

Authors: the figure caption was revised accordingly.

7.- In table 1, the value calculated is a “binding score” not a “binding energy” as is correctly stated in lines 111-112.

Authors: the term binding score was inserted in the caption of table 1.

Reviewer 2 Report

In this study the authors used an in silico approach to identify natural products as exoglucanase inhibitors of candida albicans and identified sesamin as potential inhibitor that was subsequently tested as bioactive in various assays.

The authors already write in the introduction “Targeting of this enzyme with sesamin can inhibit the morphogenetic switching of Candida spp. from yeast to virulent hyphae form.” without mentioning any reference. While its not clear which glucanase is exactly meant, it seems a bit weird that a virtual screening campaign was conducted from which sesamin resulted as only compound that was positively tested in various assays. One could get the impression that the in silico study was designed so that only sesamin results as hit from the virtual screen. In fact, no serious researcher uses only a single virtual hit for experimental tests.

It is not clear, why the authors selected the enzyme. Other exoglucanases should have been used as comparison for docking.

This sentence needs a reference:

On the other hand, natural products are considered important medicinal candidates for the development of drugs because of their less toxicity and side effects, low drug resistance, high bioavailability (oral intake) and have better curative effects.”

A thorough discussion of the results is missing, basically the authors only present their data in a very brief manner. Remarkably, the authors have not conducted a literature search on sesamin and candida in PubMed, otherwise they would have mentioned other studies in which sesamin has been tested against Candida, for example a report in which sesamin has been reported as antifungal however with a much lower MIC of 256 ug/ml (https://pubmed.ncbi.nlm.nih.gov/30549887/). In another study (https://pubmed.ncbi.nlm.nih.gov/16462055/) sesamin did not show any antifungal properties. The authors should discuss the discrepancy of these findings with their data. In another study it was also hypothesized that sesamin and fatty acid components of sesame oil are involved in its antifungal activity (https://pubmed.ncbi.nlm.nih.gov/24057219/).

Therefore, the authors are requested to compare their data with previous studies of relevance.

Details of the docking procedure (box size, box location, number of docking runs, non-default docking parameters) are missing. How the active site was discovered/selected?

Why the authors used fluconazole as reference ligand? The drug is known to target 14alpha demethylase. Therefore it does not make sense to use it for comparison.

Just looking on the docking score is not a good choice for selecting hits. Sesamin is much larger than for example luteolin, which has also a good score. Therefore score normalisation should have been applied (score/number of heavy atoms).

The statement “being able to form π-π stacking with Phe229, Tyr255, and Phe258 (Figure 3)” is wrong, as the part of sesamin interacting with Phe229 is not aromatic.

Is is not clear how the 3D structures were generated. It is unlikely that 3D structures can be obtained from PubMed, I guess the authors meant PubChem, because one can download 3D structures there. Or did the authors searched for the structures and then built the structures using a molecular editor?

He overall quality of the figures can be improved significantly. In general, the captions are too brief and should include a better description of what is shown in the figures.

The molecular interactions in figures 3 and 4 are hardly to see and the resolution is poor. Please use one plot with high resolution where only the information that you want to share is included. The captions should explain the dotted lines and there are other artifacts in the figures (small white spheres?) that should be removed.

The two plots in figure 5 do not have the same height and the colors for the axes are different. You can easily use qtgrace or xmgrace to generate one figure with two neighboured plots.

It is not clear what section is what in figure 8. A brief description of the concentrations used should be included in the caption.

Figure 9 misses A and B and a better explanation, e.g. which line is the control and which the one in presence of the compound.

The MIC and MFC data for fluconazole and amphothericin B should be compared with literature data.

The overall text quality is ok, but can still be improved significantly. The authors should check the manuscript for further improvements such as

a) remove “good” from “good promising approach” (p. 2, line 79)

b) remove “On the other hand,” (p. 2, line 80)

c) remove “experiment” in “molecular dynamics (MD) simulation experiments” (p. 2, line 86)

d) “Figure 6. Ligand monitored in the course of the MD run.” should be better “Protein-ligand interactions monitored throughout the MD simulation”.

These are just a few examples of problematic parts, please check the whole text carefully.

Overall, the manuscript is of interest, although I have doubts that the study was really carried out as described. Importantly, there are serious flaws that need to be addressed by the authors. It is not acceptable that the authors do not cite relevant literature in which sesamin has been tested for antifungal properties but with different results (low or no activity). The quality of the figures is poor and the authors should spend much more time to adequately present and discuss their data.

Author Response

In this study the authors used an in silico approach to identify natural products as exoglucanase inhibitors of candida albicans and identified sesamin as potential inhibitor that was subsequently tested as bioactive in various assays.

The authors already write in the introduction “Targeting of this enzyme with sesamin can inhibit the morphogenetic switching of Candida spp. from yeast to virulent hyphae form.” without mentioning any reference. While its not clear which glucanase is exactly meant, it seems a bit weird that a virtual screening campaign was conducted from which sesamin resulted as only compound that was positively tested in various assays. One could get the impression that the in silico study was designed so that only sesamin results as hit from the virtual screen. In fact, no serious researcher uses only a single virtual hit for experimental tests.

Authors: references have been added accordingly. We agree with the referee that usually employing a virtual screening campaign some computational hits are tested. This is true for large chemical databases. In the current study, we used a limited number of compounds in the library and so we tested only the best predicted hit for the mentioned reasons, and for saving materials considering the limited funds dedicated for this work. Further investigation of different compounds can be carried out in the next future.

It is not clear, why the authors selected the enzyme. Other exoglucanases should have been used as comparison for docking.

This sentence needs a reference:

“On the other hand, natural products are considered important medicinal candidates for the development of drugs because of their less toxicity and side effects, low drug resistance, high bioavailability (oral intake) and have better curative effects.”

Authors: a paragraph has been added in the results and discussion about the choice of the selected enzyme. The related suggested references have been added.

A thorough discussion of the results is missing, basically the authors only present their data in a very brief manner. Remarkably, the authors have not conducted a literature search on sesamin and candida in PubMed, otherwise they would have mentioned other studies in which sesamin has been tested against Candida, for example a report in which sesamin has been reported as antifungal however with a much lower MIC of 256 ug/ml (https://pubmed.ncbi.nlm.nih.gov/30549887/). In another study (https://pubmed.ncbi.nlm.nih.gov/16462055/) sesamin did not show any antifungal properties. The authors should discuss the discrepancy of these findings with their data. In another study it was also hypothesized that sesamin and fatty acid components of sesame oil are involved in its antifungal activity (https://pubmed.ncbi.nlm.nih.gov/24057219/).

Therefore, the authors are requested to compare their data with previous studies of relevance.

Authors: (https://pubmed.ncbi.nlm.nih.gov/30549887/) This reference is mentioned in our manuscript having reference number [24] at line number 96 and in the conclusion section, named as “Agbo, J. B., Mpetga, J. D., Bikanga, R., Tchuenguem, R. T., Tsafack, R. B., Awouafack, M. D., ... & Tane, P. (2017). A new benzophenanthridine alkaloid from Caloncoba glauca. Natural Product Communications12(3), 1934578X1701200314”.

In this study authors have reported the 256 μg/mL, a higher MIC value against Candida albicans, but our study has reported the 16 μg/mL, lower MIC values against different strains of Candida spp. both by Disc diffusion method and Broth microdilution method (according to protocol mentioned by CLSI guidelines, M27S4). Our study also confirmed the antifungal activity of sesamin both by in-silico and in-vitro work.

(https://pubmed.ncbi.nlm.nih.gov/16462055/) we have read this reference, the study does not mention about the MIC value of sesamin against Candida spp, so we have not included this reference in our manuscript.

(https://pubmed.ncbi.nlm.nih.gov/24057219/) we have added this reference in our manuscript having reference number [15] , named as “Effect of Edible Sesame Oil on Growth of Clinical Isolates of Candida albicans Toshiko Ogawa, RN, PHN, MMS1 , Junko Nishio, MT, CTIAC1 , and Shinobu Okada, RN, PHN, PhD”

In this study, authors had reported the antifungal activity of sesame oil against Candida albicans by studying the inhibitory effect of different concentration of sesame oil on mycelial forms and yeast forms. Our study is also describing the antifungal activity of sesamin (present in sesame oil), both by in silico and in vitro method by targeting the mycelial or hyphae form of Candida spp. involved in the pathogenesis of host tissues, thus by inhibiting morphogenetic switching.

Details of the docking procedure (box size, box location, number of docking runs, non-default docking parameters) are missing. How the active site was discovered/selected?

Authors: Details of docking procedure have been added. The selected binding site was the site derived from the crystallization studies in which a ligand is present. Accordingly, we centered the grid on the region in which the binding site is located being sure that the whole volume of the binding site was covered by the grid for docking studies.

Why the authors used fluconazole as reference ligand? The drug is known to target 14alpha demethylase. Therefore it does not make sense to use it for comparison.

Authors: Fluconazole is defined as first line of antifungal drug, used in our study because it is a popular drug and drug of choice in India and easily available in the market and also effective against strains found in India. It is used in our study to check the efficacy of drug not its mode of action. Accordingly, was used as reference drug for the antifungal activity, and considering the comment of the referee, we delete the discussion about the docking of fluconazole since as argued from the referee the target of fluconazole is different.

Just looking on the docking score is not a good choice for selecting hits. Sesamin is much larger than for example luteolin, which has also a good score. Therefore score normalisation should have been applied (score/number of heavy atoms).

Authors: our study focusses on the docking score for the selection of hits. In our study, sesamin was found to have lower binding energy score considering the selected enzyme. Furthermore, due to the small differences in the size of the most promising compounds the calculation of ligand efficiency (score/number of heavy atoms) did not bring any modification to the docking output.

The statement “being able to form π-π stacking with Phe229, Tyr255, and Phe258 (Figure 3)” is wrong, as the part of sesamin interacting with Phe229 is not aromatic.

Authors: we modified the figure 3 in order to better understand the pattern of interaction. Accordingly, the interaction with Phe229 and Tyr255 was a π-π stacking, while with Phe 258 is a π-alkyl contact. The current sentence has been modified as follows: being able to form π-π stacking with Phe229 and Tyr255, while we detected a π-alkyl interaction with Phe258.

Is is not clear how the 3D structures were generated. It is unlikely that 3D structures can be obtained from PubMed, I guess the authors meant PubChem, because one can download 3D structures there. Or did the authors searched for the structures and then built the structures using a molecular editor?

Authors: we apologize for the mistake. We have corrected PubMed in PubChem.

He overall quality of the figures can be improved significantly. In general, the captions are too brief and should include a better description of what is shown in the figures.

The molecular interactions in figures 3 and 4 are hardly to see and the resolution is poor. Please use one plot with high resolution where only the information that you want to share is included. The captions should explain the dotted lines and there are other artifacts in the figures (small white spheres?) that should be removed.

Authors: as previously mentioned the figure has been replace in order to provide a better viewing.

The two plots in figure 5 do not have the same height and the colors for the axes are different. You can easily use qtgrace or xmgrace to generate one figure with two neighboured plots.

Authors: the figures were done with the tools implemented in Desmond software considering the MD simulation, we used from several years this method of representation. Considering the amount of data for generating the plots, it is counterproductive to use a different software to plot the output of MD simulation in terms of RMSD and RMSF. However, to address this request we used the same tool implemented in Desmond for both graphs, replacing the original figure.

It is not clear what section is what in figure 8. A brief description of the concentrations used should be included in the caption.

Authors: Fig 8 caption was revised accordingly.

Figure 9 misses A and B and a better explanation, e.g. which line is the control and which the one in presence of the compound.

Authors: Fig 9 was revised accordingly.

The MIC and MFC data for fluconazole and amphothericin B should be compared with literature data.

Authors: details have been added accordingly

The overall text quality is ok, but can still be improved significantly. The authors should check the manuscript for further improvements such as

  1. a) remove “good” from “good promising approach” (p. 2, line 79)
  2. b) remove “On the other hand,” (p. 2, line 80)
  3. c) remove “experiment” in “molecular dynamics (MD) simulation experiments” (p. 2, line 86)
  4. d) “Figure 6. Ligand monitored in the course of the MD run.” should be better “Protein-ligand interactions monitored throughout the MD simulation”.

These are just a few examples of problematic parts, please check the whole text carefully.

Authors: examples have been corrected and a careful check on the manuscript was performed to avoid typo and grammar errors.

Overall, the manuscript is of interest, although I have doubts that the study was really carried out as described. Importantly, there are serious flaws that need to be addressed by the authors. It is not acceptable that the authors do not cite relevant literature in which sesamin has been tested for antifungal properties but with different results (low or no activity). The quality of the figures is poor and the authors should spend much more time to adequately present and discuss their data.

Authors: we thank the referee for the suggestion. Accordingly, the papers have been cited, and the figures have been replaced for improving the overall quality of the paper.

Reviewer 3 Report

The paper presents an integrated in silico and biological screening platform for the identification of sesamin as a potent antifungal agent. Despite the performed detailed investigation, a variety of applied methods and the quality of the obtained results, the paper needs some additional efforts before to be considered for publishing in such an esteemed journal as Molecules.

Although promising results obtained about sesamin, showing its potential to be considered as an antifungal agent, I find this investigation unsatisfactory without some additional comparison with any of other phytocompounds presented in Table 1. Despite the fact that sesamin has shown the lowest binding energy after molecular docking, it seems too me too restrictive to subject only this compound to further analysis. I would suggest at least 2 or 3 more compounds, representatives of different classes of phytocompounds and showing also very promising binding energies as shown in Table 1, to be analyzed in the way as it has been done for sesamin.

In addition, from the paper it seems that there should be some “Supplementary Materials: The following supporting information can be downloaded at: www.mdpi.com/xxx/s1, Figure S1: title; Table S1: title; Video S1: title.”, while such are not mentioned in the text and, in fact, are not provided.

Author Response

The paper presents an integrated in silico and biological screening platform for the identification of sesamin as a potent antifungal agent. Despite the performed detailed investigation, a variety of applied methods and the quality of the obtained results, the paper needs some additional efforts before to be considered for publishing in such an esteemed journal as Molecules.

Although promising results obtained about sesamin, showing its potential to be considered as an antifungal agent, I find this investigation unsatisfactory without some additional comparison with any of other phytocompounds presented in Table 1. Despite the fact that sesamin has shown the lowest binding energy after molecular docking, it seems too me too restrictive to subject only this compound to further analysis. I would suggest at least 2 or 3 more compounds, representatives of different classes of phytocompounds and showing also very promising binding energies as shown in Table 1, to be analyzed in the way as it has been done for sesamin.

Authors: We agree with the referee that usually employing a virtual screening campaign some computational hits are tested. This is true for large chemical databases. In the current study, we used a limited number of compounds in the library and so we tested only the best predicted hit for the mentioned reasons, and for saving materials considering the limited funds dedicated for this work. Further investigation of different compounds can be carried out in the next future. Furthermore, we added some information about other promising compounds now in Table 2.

Reviewer 4 Report

The authors performed in silico and in vitro studies aiming to find a compound targeting the enzyme 1,3-β-D-glucan synthase. The manuscript is within the Molecules scope; however, it will be suitable for publication only after some aspects to be considered. These issues are listed and discussed in the following topics:

(1) I suggest the inclusion of the molecular target name in the manuscript title. 

(2) Introduction contains two paragraphs and the first paragraph is so long. Please rewrite aiming to make it more compressive and well-structured.

(3) Is unclear the 40 compounds choice. This aspect needs to be clearly justified and discussed.

(4) Another point to be reported by the authors is the source of Sesamin. Was it purchased or isolated? if it was isolated are there available spectra data aiming the compound identity confirmations? 

(5) In unclear the choice of fluconazole as a ligand in docking in 1,3-β-D-glucan synthase. The main accepted target for fluconazole is 14-a-demethylase

(6) I suggest present the in silico results in crescent complexity order, starting with in silico ADMET properties, docking studies, and molecular dynamics. This order is more logical and possibly better represents the order in which the results were obtained.

(6) The authors opted by the description of their results together with discussion, however, the results are poorly discussed or many times no discussed. No references were cited in the section “results and discussion”!

(8) The confirmation of synergism involves the construction of an isobologram. The results about this are preliminary and need proof by using suitable statistical methods.

(9) Please see the use of italics in Latin names

Author Response

The authors performed in silico and in vitro studies aiming to find a compound targeting the enzyme 1,3-β-D-glucan synthase. The manuscript is within the Molecules scope; however, it will be suitable for publication only after some aspects to be considered. These issues are listed and discussed in the following topics:

(1) I suggest the inclusion of the molecular target name in the manuscript title. 

Authors: considering that we investigated the antifungal activity of sesamin proposing a possible target but unfortunately, due to the time for the revision and the difficult to have an efficient test considering the selected target, we prefer to maintain the original title.

(2) Introduction contains two paragraphs and the first paragraph is so long. Please rewrite aiming to make it more compressive and well-structured.

Authors: the corrections were inserted in the revised version.

(3) Is unclear the 40 compounds choice. This aspect needs to be clearly justified and discussed.

Authors: the selection of the compounds was better stated in the revised version. The phytocompounds were selected based on Lipinski Rule of Five.

(4) Another point to be reported by the authors is the source of Sesamin. Was it purchased or isolated? if it was isolated are there available spectra data aiming the compound identity confirmations?

Authors: Sesamin was purchased from Combi Blocks (Purity 98%), this point is now mentioned in manuscript in the materials and methods section. COMBI-BLOCKS is a research-based manufacturer and worldwide supplier of lab reagents.

(5) In unclear the choice of fluconazole as a ligand in docking in 1,3-β-D-glucan synthase. The main accepted target for fluconazole is 14-a-demethylase

Authors: Fluconazole is defined as first line of antifungal drug, used in our study because it is a popular drug and drug of choice in India and easily available in the market and also effective against strains found in India. It is used in our study to check the efficacy of drug not its mode of action. Accordingly, was used as reference drug for the antifungal activity, and considering the comment of the referee, we delete the discussion about the docking of fluconazole since as argued from the referee the target of fluconazole is different.

(6) I suggest present the in silico results in crescent complexity order, starting with in silico ADMET properties, docking studies, and molecular dynamics. This order is more logical and possibly better represents the order in which the results were obtained.

Authors: in Table 1 are reported the parameters for taking into account the Lipinski rule of 5 that is the parameter used for selecting the phytochemicals. After that, the in depth investigation about the admet properties was performed for the best computational hit sesamin using different tools for providing a comprehensive assessment of the physchem profile of sesamin..

(6) The authors opted by the description of their results together with discussion, however, the results are poorly discussed or many times no discussed. No references were cited in the section “results and discussion”!

Authors: results and discussion section has been revised in order to provide a better discussion of the results.

(8) The confirmation of synergism involves the construction of an isobologram. The results about this are preliminary and need proof by using suitable statistical methods.

Authors: We have studied literature data focussing on checkerboard microdilution assay/ synergistic study describing antifungal activity of compound with commercially available drug. The results are described in same way that we have shown in our manuscript by calculating fractional inhibitory concentration index (FICI). The experiments are carried out as done in the manner as done in all laboratories and results are described in the same way as is reported in different research articles worldwide.

(9) Please see the use of italics in Latin names

Authors: correction done

Round 2

Reviewer 1 Report

The manuscript was corrected taking into account the most of the suggestions. However, the key points, such as the characterization of only one compound, or the lack of enzyme inhibition studies, were not covered. The authors make a statement at this respect, and the problems described about enzyme activity assays are understandable. However, to perform the characterization of only one compound, that was selected from a molecular docking protocol, generates a study with limited scientific soundness, as was suggested, it is necessary to include the characterization (in silico and biological activity) of more compounds to improve the manuscript and to support the global strategy followed.

Author Response

The manuscript was corrected taking into account the most of the suggestions. However, the key points, such as the characterization of only one compound, or the lack of enzyme inhibition studies, were not covered. The authors make a statement at this respect, and the problems described about enzyme activity assays are understandable. However, to perform the characterization of only one compound, that was selected from a molecular docking protocol, generates a study with limited scientific soundness, as was suggested, it is necessary to include the characterization (in silico and biological activity) of more compounds to improve the manuscript and to support the global strategy followed.

Authors: In this study, we got good results with sesamin in in-silico analysis that is the reason why we further performed experiments to determine its antifungal activity against different Candida spp. However based on the suggestion provided from reviewer 1, we have checked Adsorption, distribution, metabolism and excretion (ADME) profile of pinoresinol, caflanone, herbacetin and sesamin and we have found best results with sesamin. The results of ADME profile have now been added in the manuscript (Table 5)

Reviewer 2 Report

Issues have been addressed, the manuscript is suited for publication.

Author Response

Issues have been addressed, the manuscript is suited for publication.

Authors: we thank the referee for the positive evaluation of the revised version of the paper.

Reviewer 3 Report

I accept the authors consideration about the questions set by me in the review and appreciate the Table 2 inclusion. The paper quality is significantly improved in this stage, thus I would suggest the manuscript to be accepted for publication.

Author Response

I accept the authors consideration about the questions set by me in the review and appreciate the Table 2 inclusion. The paper quality is significantly improved in this stage, thus I would suggest the manuscript to be accepted for publication.

Authors: we thank the referee for the positive evaluation of the revised version of the paper.

Reviewer 4 Report

(1) figure 2 is unnecessary. The sesamin structure is showed in others parts of the manuscript

(2) in some figures are missing the identification of the parts (a), (b)…

(3) without an enzymatic assay, the results are very preliminary for publication in Molecules

(4)  sesamin effect on candida species was previously reported, so that the results have only incremental value. In addition, the authors obtained some discrepant results.

Author Response

(1) figure 2 is unnecessary. The sesamin structure is showed in others parts of the manuscript

Authors: As per suggestions of the reviewer 4, we have removed chemical structure of sesamin (figure 2 in the previous study) from the manuscript.

(2) in some figures are missing the identification of the parts (a), (b)…

Authors: we thank the reviewer for the careful reading. Accordingly, each figure that present more than one panel has been correctly labelled with letters as reqiested

(3) without an enzymatic assay, the results are very preliminary for publication in Molecules

Authors: As already mentioned in the manuscript, the investigations are based on in silico analysis followed by preliminary antifungal assays that include determination of minimal inhibitory conc. of sesamin with four clinically relevant Candida strains, spot assays and synergistic drug assays.   

For confirmatory enzymatic assays, we might go for isolating the enzyme from Candida spp. which can be planned in the near future.

However in this particular study, we are reporting good antifungal activity (MIC: 16 µg/mL which is much better than reported previously i.e. 256 μg/mL (Reference no. 24 Agbo, J. et al. A New Benzophenanthridine Alkaloid from Caloncoba glauca. Nat Prod Commun 2017, 12, 367-368).

(4)  sesamin effect on candida species was previously reported, so that the results have only incremental value. In addition, the authors obtained some discrepant results.

Authors: Weak antifungal activity of sesamin has been reported against only one fungal strain Candida albicans in 2017 (reference no. 24 Agbo et al.). We are reporting significant antifungal activity against four clinically relevant Candida strains (which also include one strain resistant to fluconazole-C. krusei). Besides our study has shown good synergistic antifungal activity of sesamin with the most commonly used drug in India (Fluconazole). The results give an indication that we can significantly reduce the toxicity due to conventional drugs, making sesamin a safe and promising alternative.

Round 3

Reviewer 1 Report

To include only ADME predictions from three more compounds is not enough to increase the scientific soundness of the manuscript. In general, the values obtained are very similar, or at least, nothing is mentioned in a specific manner about the difference among them to state that sesamin is better. Furthermore, these results do not exclude the necessity, to probe experimentally and to perform MD studies, of these three compounds because predicted ADME parameters are not necessarily related with biological activity. Additionally, the selection criteria was docking score and not ADME properties. In conclusion, this modification does not increase the quality of the manuscript and the concerns continue being the same mentioned in 2nd round revision. 

Reviewer 4 Report

I believe that the absence of enzymatic assays, the publications about the activity of sesamin available in the literature and the large amount of in silico data with a small amount of in vitro or in vivo experiments are limitations for the publication in a journal as Molecules.